# Beyond Interpolation: Extrapolative Reasoning with Reinforcement Learning and Graph Neural Networks

**Niccolò Grillo**[1*], **Andrea Toccaceli**[1*], **Joël Mathys**[2], **Benjamin Estermann**[2],
**Stefania Fresca**[1], **Roger Wattenhofer**[2]

[1]MOX - Department of Mathematics, Politecnico di Milano, Italy
[2]ETH Zurich, Switzerland
{niccolo.grillo, andrea.toccaceli}@mail.polimi.it, stefania.fresca@polimi.it, {jmathys, estermann, wattenhofer}@ethz.ch

## Abstract

Despite incredible progress, many neural architectures fail to properly generalize beyond their training distribution. As such, learning to reason in a correct and generalizable way is one of the current fundamental challenges in machine learning. In this respect, logic puzzles provide a great testbed, as we can fully understand and control the learning environment. Thus, they allow to evaluate performance on previously unseen, larger and more difficult puzzles that follow the same underlying rules. Since traditional approaches often struggle to represent such scalable logical structures, we propose to model these puzzles using a graph-based approach. Then, we investigate the key factors enabling the proposed models to learn generalizable solutions in a reinforcement learning setting. Our study focuses on the impact of the inductive bias of the architecture, different reward systems and the role of recurrent modeling in enabling sequential reasoning. Through extensive experiments, we demonstrate how these elements contribute to successful extrapolation on increasingly complex puzzles. These insights and frameworks offer a systematic way to design learning-based systems capable of generalizable reasoning beyond interpolation.

## Introduction

Neural architectures have made significant strides in various domains, yet a fundamental challenge persists: the ability to generalize effectively beyond their training distribution. This limitation is particularly evident in tasks requiring logical reasoning, where the structured nature of the problem amplifies the consequences of poor generalization. Consider a neural network trained to solve 3x3 Sudoku puzzles. While it may excel within this confined space, it often fails when presented with 4x4 or 9x9 grids, despite the underlying logical principles remaining the same. This toy example underscores a critical issue: many neural models achieve good performance during training, but fail to extract the fundamental logical relationships and dynamics governing the problem. The ultimate goal of these neural architectures is not to just interpolate between seen examples, but to genuinely understand, extract, and correctly reapply the underlying reasoning and knowledge. The key lies in developing systems that

can comprehend and reason with core logical structures, enabling them to apply this knowledge to novel, more complex scenarios. This level of generalization - moving beyond interpolation to true logical understanding - is essential for creating future machine learning systems capable of robust reasoning across diverse contexts, including those that go beyond their training experiences.

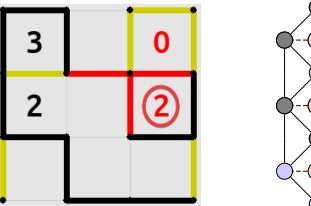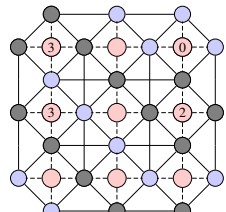

Figure 1: We focus on logic puzzles of varying sizes in order to systematically evaluate the ability of neural architectures to extrapolate beyond the seen training data. By modelling the problem instances through a unifying graph framework, we can naturally encompass and evaluate on instances where generalization capabilities are required.

Logic puzzles provide an ideal testbed for addressing the generalization challenge in neural architectures. These puzzles offer a unique environment with clear rules and scalability. This allows us to systematically investigate a model's ability to reason beyond its training distribution by applying it to larger puzzle sizes. The inherent advantage of evaluating generalization through extrapolation is that larger puzzle configurations clearly lie outside the training distribution while the underlying rules stay the same. Therefore, we desire models that cannot only learn the correct principles from simpler scenarios, but also be applied to entirely new contexts. By controlling the difficulty and structure of puzzles, we can create such scenarios by testing whether a model has truly grasped the underlying logical principles or merely memorized patterns. Moreover, the unambiguous nature of puzzle solutions enables objective and exact measurement of performance.

We choose to study this problem using Reinforcement Learning (RL), as it offers distinct advantages, particu-

---
*These authors contributed equally.

larly in modeling the sequential decision-making process inherent in puzzle-solving. This approach mirrors human problem-solving strategies, allowing the agent to learn and refine its approach over time. The RL framework balances exploration of new strategies with exploitation of known effective approaches, which can be beneficial for discovering generalizable solutions. Supervised generalization benchmarks such as CLRS (Velickovic et al. 2022) often rely on intermediate hints to match the ground-truth algorithm during the training process. In the RL framework, on the other hand, the agent is free to pursue any algorithm that ensures high reward. Moreover, RL's customizable reward structure enables us to encourage true understanding over mere memorization by rewarding not just correct solutions, but also the application of sound logical principles.

An inherent challenge with testing generalization is the need for the neural agent to be able to handle logic puzzles of different sizes. For many neural architectures, it is not possible to even represent instances of different sizes, or they struggle to properly adapt as the scale of the problem changes. This limitation severely constrains their ability to generalize to larger, more complex puzzles. To address this challenge, we propose to represent logic puzzles as graphs. In our graph-based modeling, puzzle elements become nodes, while the relationships and constraints between these elements are represented as edges, providing a level of abstraction as depicted in Figure 1. This approach allows us to apply Graph Neural Networks (GNNs), which can naturally adapt to various puzzle sizes while maintaining a consistent structure. As puzzle size increases, the graph simply expands without fundamentally altering the underlying representation.

In this paper, we focus on how to best model and develop techniques for neural architectures that use the proposed graph structures in order to solve logic puzzles. More precisely, we propose to tackle the problem using a multi-agent context and utilize GNNs in order to directly exploit the topological structure of the puzzles. This approach allows us to represent both the puzzle structure and the problem-solving agents within a unified framework. In addition, we evaluate whether the inductive biases of GNNs provide advantages over a more general architecture such as transformers. Then, we explore what factors impact on the ability to generalize and extrapolate. Here, we focus on two key areas: the design of reward systems and the role of recurrent modeling in sequential reasoning.

We summarize our contributions as follows:

- Introduction of a novel graph-based evaluation framework for logic puzzles with a focus on scaling to varying problem sizes, specifically designed to test extrapolation beyond the training distribution.
- Proposal of a multi-agent reinforcement learning approach using Graph Neural Networks, which models

---

puzzle-solving as a collaborative task and enables learning of generalizable strategies across different puzzle sizes and complexities.

- Insights into the factors that influence generalization in logical reasoning tasks. This includes the inductive bias of the architecture, the role of reward design, and sequential decision-making mechanisms.
- Demonstration of our approach's effectiveness through extensive experiments on a range of logic puzzles, showing improved performance and generalization capabilities compared to existing methods.

## Related Work

**Reasoning and Generalization**  Over the years the study of reasoning with learning based approaches has often focused on the domain of games such as chess, shogi and Go (Lai 2015; Silver et al. 2016, 2017, 2018), Poker (Dahl 2001; Heinrich and Silver 2016; Steinberger 2019; Zhao et al. 2022) or board games (Ghory et al. 2004; Szita 2012; Xenou, Chalkiadakis, and Afantenos 2019; Perolat et al. 2022). While these mainly focus on correct play or in-distribution performance, the CLRS Algorithmic Reasoning Benchmark introduced by Velickovic et al. (2022) puts emphasis on generalizable reasoning. It consists of a diverse collection of well known algorithms collected from the textbook "Introduction to Algorithms" by Cormen et al. (2022) providing a resource to assess algorithmic reasoning for learning based approaches. Moreover, there exists a CLRS-Text (Markeeva et al. 2024) to better assess the reasoning capabilities from a language perspective as well. Abbe et al. (2024) provide a more theoretically supported view on the generalization performance of some classes of neural networks, trained with SGD. They specifically focus on Boolean functions, learned in a supervised training regime.

**Graph Neural Networks**  First introduced by the works of Scarselli et al. (2008), Graph Neural Networks have seen a recent emergence through a variety of new architecture types and applications (Kipf and Welling 2017a; Xu et al. 2019; Veličković et al. 2018). The graph-based representation underlying these models is particularly powerful as it provides a natural framework for capturing relational information and structural dependencies between entities. This has made GNNs especially interesting for tackling combinatorial optimization problems (Dai et al. 2018; Cappart et al. 2022; Tönshoff et al. 2022) and reasoning tasks that require understanding relationships between multiple elements (Battaglia et al. 2018). A key advantage of graph-based approaches is their capability to handle problems of varying sizes and complexities. One specific research direction focuses how these models generalize across different problem instances and sizes (Xu et al. 2021; Schwarzschild et al. 2021), with benchmarks like CLRS (Velickovic et al. 2022) providing a more systematic evaluation frameworks for assessing algorithmic reasoning capabilities with a focus on size generalization. This in turn has sparked more investigations into developing appropriate tools and architectures for such reasoning (Ibarz et al. 2022; Numeroso, Bacciu, and

---

The code will be made available as part of https://github.com/ETH-DISCO/rlp

Veličković 2023; Minder et al. 2023; Mahdavi et al. 2023; Bohde et al. 2024; Müller et al. 2024).

## Preliminaries

### PUZZLES Benchmark

The PUZZLES benchmark, introduced by Estermann et al. (2024) is a comprehensive testing environment that aims to evaluate and enhance the algorithmic reasoning capabilities of reinforcement learning (RL) agents. It is centered around Simon Tatham's Portable Puzzle Collection (Tatham 2004) and encompasses 40 diverse logic puzzles that range in complexity and configuration, providing a rich ground for assessing RL algorithms' performance in logical and algorithmic reasoning tasks. In the PUZZLES benchmark, RL agents can interact with the puzzles using either a visual representation (e.g., images of the game) or a discretized representation in the form of tabular data describing the current state of the puzzle. The puzzle instances are configurable in terms of difficulty and size, allowing researchers to test different RL algorithms' scalability and generalization abilities. Additionally, PUZZLES is integrated into the widely-used Gymnasium framework.

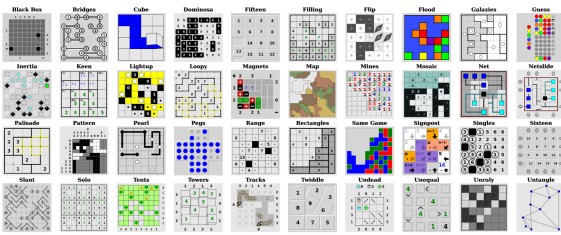

Figure 2: Some Example puzzles of the PUZZLES library, inspired by the collection of Simon Tatham.

PUZZLES allows for the use of different observation spaces, including pixel-based inputs similar to those used in classic RL benchmarks like Atari (Bellemare et al. 2013), as well as discrete state representations that can be more suitable for logic-intensive tasks. Even though, the PUZZLES framework already allows to test generalization across different puzzles sizes in principle, preliminary results show that the proposed RL baselines struggle a lot in that domain. One current limitation is that the puzzle interface used for these test do not allow for an easy or natural way of adjusting to larger puzzle instances. In our work, we propose to overcome this issue by using a graph-based representation interface.

### Graph Neural Networks

Graph Neural Networks (GNNs) are a class of neural networks specifically designed to operate on graph-structured data. Traditional neural architectures, such as Convolutional Neural Networks or Recurrent Neural Networks are tailored for processing image or sequential data. In contrast, GNNs are specifically designed to handle relational data by incorporating important symmetries of the data within its architecture.

A graph $G = (V, E)$ consists of $V$, the set of nodes and $E$, the set of edges. Each node $v \in V$ and each edge $e = (u, v) \in E$ may have associated feature vectors $h_v$ and $h_e$, respectively. Most common GNNs operate on the principle of message-passing which involves iterative neighborhood aggregation and node updates. Importantly, the mechanism is shared across all nodes in the graph, which allows GNNs to be applied to graphs of varying sizes, regardless of the number of nodes or edges. We follow the notion of Xu et al. (2019) to express the $t$-th layer as follows:

$$
\begin{aligned}
a_v^t &= \text{AGGREGATE}^t(\{\{h_u^t \mid u \in N(v)\}\}) \\
h_v^{t+1} &= \text{COMBINE}^t(h_v^t, a_v^t).
\end{aligned}
$$

The original input features of the nodes are defined as $h_v^0$ and messages from the neighborhood $N(u)$ are aggregated and then combined with the previous state. In practice, we rely on parameterized functions $\psi$ and $\phi$ and use a permutation invariant aggregator $\bigoplus$ such as sum.

$$
\mathbf{h}_v^{t+1} = \phi\left(\mathbf{h}_v^t, \bigoplus_{u \in \mathcal{N}(v)} \psi\left(\mathbf{h}_u^k, \mathbf{h}_v^k, \mathbf{h}_{uv}\right)\right)
$$

With each added layer or round of message passing, the receptive field of the nodes is increased. After $k$ rounds, a node's representation has been influenced by its k-hop neighborhood. This relationship between the number of rounds and the receptive field size is crucial for capturing local and global graph structures effectively. Moreover, we usually distinguish between node and graph level prediction tasks:

$$
\begin{aligned}
y_v &= \varphi\left(h_v^k\right) \\
y_G &= \varphi_G\left(\bigoplus_{u \in V} \varphi\left(h_u^k\right)\right).
\end{aligned}
$$

For node prediction $y_v$, the output of each node is typically a transformation of the last embedding after $k$ rounds of message passing. Whereas for graph level prediction $y_G$, we usually apply a graph-pooling operation to aggregate all final node embeddings into a graph representation.

### Reinforcement Learning

Reinforcement learning (RL) focuses on training agents to make sequential decisions in an environment to maximize cumulative rewards. An RL agent learns through trial and error by interacting with the environment: observing the current state, taking an action, and receiving a reward. This process is repeated, with the agent aiming to learn a policy $\pi(a|s)$ that maximizes cumulative rewards over time. A common way to formalize an RL problem is as a Markov Decision Process (MDP), defined by states $S$, actions $A$, transition probabilities $P$, rewards $R$, and a discount factor $\gamma$. The agent's goal is to find a policy that maximizes the expected cumulative discounted reward:

$$
\mathbb{E}_\pi\left[\sum_{t=0}^{\infty} \gamma^t R(s_t, a_t, s_{t+1})\right].
$$

In this work, we use a model-free RL approach with Proximal Policy Optimization (PPO) (Schulman et al. 2017) to train agents on a subset of the PUZZLES benchmark. PPO is a popular model-free algorithm known for its stability and efficiency in finding effective policies.

## Methodology

### Modeling Puzzles as Graphs

The PUZZLES benchmark (Estermann et al. 2024) provides a starting point for the selection of appropriate logic puzzles. While this benchmark already provides access to varying difficulties and puzzle sizes, there are a few details that make it challenging to study size generalization directly. The interface only provides the pixel observations or a discretized tabular view of the puzzle. This makes the development of models which can incorporate such a representation well when the size is varying challenging. Another aspect is that not all puzzles are equally suitable for the study of extrapolation. Indeed, some have interactive elements which make them more complex by design, while others rely on novel elements when increased in size. This adds another challenge of value generalization, even if the rules were learned correctly. For example Sudoku introduces new actions and elements in the form of additional numbers in larger instances. We aim to select a subset of puzzles which is large and diverse enough, but at the same time tries to decouple unnecessary complexities for the goal of evaluating extrapolation. Additionally, PUZZLES proposes an action space that involves moving a cursor around the playing field to then change the game state at its position. While this helps in providing a consistent action space for all puzzles, it adds an additional layer of complexity to the learning process. We therefore propose to focus on the following criteria:

1. *Action Space*
   The ability to describe the game as a collection of atomic cells that together represent the full state of the game.

2. *Independence*
   Each cell can take an action independently of the actions of other cells. The resulting configuration does not necessarily have to be valid.

3. *Solvability*
   The games have no point of no return (e.g., chess). At every time step, from the current state of the game, it is always possible to solve the game.

4. *Complete Information*
   There are no interactive or stochastic elements in the game. Given just the initial state of the game, it is possible to derive its solution.

5. *Value Generalization*
   The game limits it's exposure to new concepts (new elements, increased value ranges) as it increases in size. The number of actions remains constant.

To create a strong and diverse benchmark, we select six puzzles that fit the above listed criteria: Light Up, Loopy, Mosaic, Net, Tents, and Unruly also shown in Figure 3. For an in-depth description of the puzzles and the rules we refer to the Appendix.

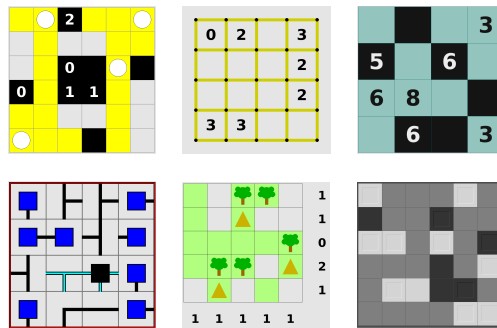

Figure 3: We provide a new graph interface in order to ease the testing for size generalization on six puzzles. From top left to bottom right: Light Up, Loopy, Mosaic, Net, Tents and Unruly.

As previously stated, in order to have an appropriate representation that can facilitate testing extrapolation on various puzzle sizes, we propose to use graphs as a unifying structure for the puzzles. By representing the puzzles as graphs, they can naturally accommodate different sizes, while preserving the local relationships of the puzzle. For each puzzle type, we provide an interface that transforms the given puzzle state into a graph. This transformation is specified by the graph topology, the types of nodes and edges, their annotated features and the allowed actions that each node can take. In general, we try to provide a very general and unified interface as not to impose design choices onto the model architectures.

Each graph representation consists of two types of nodes, which we will exemplify by the game Loopy depicted in Figure 4. The *decision-nodes*, which represent the atomic cells of a game and the *meta-nodes* which contain shared information about a collection of nodes. Each decision-node directly influences the state of the game, as it can take an action. In Loopy, every edge of the original grid is modeled as a decision-node, which determines if an edge should be present or not. The meta-nodes cannot directly alter the state of the game, instead they contain information such as a constraint or if a violation is present. In Loopy, these are the faces of the original grid. The edges of the graph usually represent neighboring cells in the original game, here adjacent edges or faces. Moreover, each node and edge is assigned a feature vector containing information about the state of the game or the direction of the edge. Note that as the puzzle is scaled to larger instances, the local graph representation remains identical across puzzle sizes. A detailed explanation on how each puzzle is modeled and details on the graph topology and action spaces is contained in the Appendix.

### Training

We follow the training process outlined in PUZZLES (Estermann et al. 2024), using PPO as a relatively simple, yet solid training algorithm. Corresponding to the graph observation, each decision node in the graph represents a possible action in the game. For each puzzle, we have a distinct set of actions associated with every decision node. The fi-

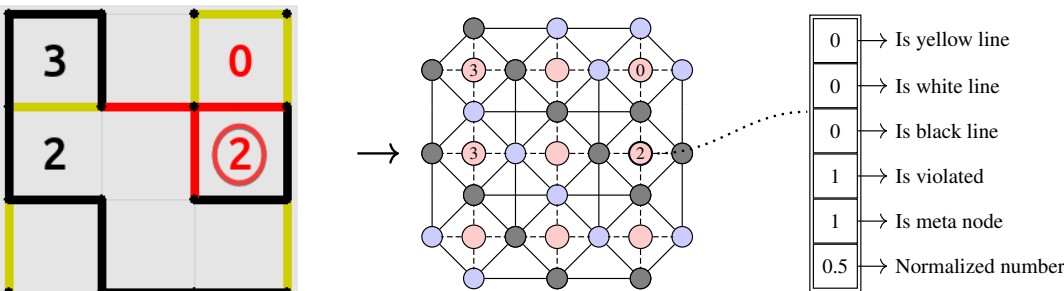

Figure 4: Illustration of the modeling of the puzzle Loopy. In this case, each decision-node (black or blue circles) corresponds to an edge of the original game grid. Each face of the game grid is represented as a meta-node (red circle), which is connected to its four adjacent decision-nodes. The nodes and edges of the graph have features, which determine the current state. The next state is determined by the collective actions of all decision-nodes. The local graph representation remains the same across all puzzle sizes.

nal action for each node is selected independently using a softmax layer. Once the actions are chosen for all nodes, they are executed simultaneously, resulting in a new state of the game. This approach differs from the cursor model used in the PUZZLES benchmark, however, it removes an additional layer of complexity and improves training efficiency. For this reason and our more dense reward scheme, we are able to drastically reduce the rollout length, leading to more efficient training.

For each combination of puzzle and presented architecture configuration we report the performance over three different model seeds. In order to offer some insight into the generalization capabilities, model selection is performed on slightly larger configurations similar to previous work (Müller et al. 2024; Jung and Ahn 2023). For model selection, we assess the performance every 100 training iterations on the validation step which contains puzzles one size larger than during training.

### Extrapolation Evaluation

The most important aspect about our benchmark is to test the capability of generalizing outside the training distribution to larger puzzles. While other metrics such as training behaviour or in-distribution-performance are of interest, we

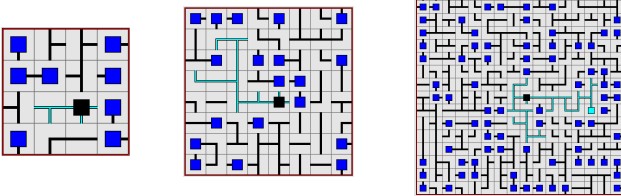

Figure 5: Illustration of testing the ability to generalize beyond the training distribution for the puzzle Net. While models only see small puzzle instances during training, the rules and logic that govern the puzzle remain the same. Therefore, during evaluation, the model is tested on puzzles that are up to 16x larger.

want to provide a comprehensive evaluation scheme for extrapolation.

To this end, we provide for each puzzle a dedicated training size and a set of larger test sizes. Unfortunately, due to the individual puzzle constraints, not all sizes can be exactly the same across the puzzles. We determine the training size of each puzzle to be the smallest size that contains a sufficient amount of unique training samples, specifically 40'000 instances. Moreover, in order to compare the extrapolation performance we aim to provide a set of sizes for each puzzle so that the overall relative increase in the sizes is similar across puzzle sizes. Each puzzle contains six test sets. One that matches the training size, the next two bigger puzzles sizes +1 and +2 as well as puzzles that are factors x4, x9, x16 larger. For each test set we test on a set of 50 different puzzles and we report the number of fully solved instances. For more details we refer to Appendix.

### Architectures

We distinguish between two different approaches to solve the puzzle, a recurrent and a state-less mode. In the recurrent mode, at each step $t$ of the puzzle solving, the model gets as input the current state of the puzzle $x^t$ as well as the previous hidden state of the model $h^t$. Then, these embeddings are given to a processor module which can be either a GNN module or a transformer based architecture. Finally, the output of the model is $o^t$, the action for each decision-node of the puzzle and the next hidden state which is derived using a GRU unit helping to stabilize the computation over longer sequences. Whereas in the state-less mode, no hidden states $h^t$ are computed and only $x^t$, the current state of the puzzle, is provided as the sole input:

$$
\begin{aligned}
z^t &= \phi_1(x^t \| h^t) \\
p^t &= z^t + \text{Processor}(z^t) \\
o^t &= \phi_2(p^t) \\
h^{t+1} &= \text{GRU}(h^t, p^t).
\end{aligned}
$$

We use two different processors, a graph and a transformer baseline. The graph baseline uses a GCN (Kipf and Welling

2017b) which executes three rounds of message passing on the provided graph structure. The transformer baseline uses an encoder transformer (Vaswani et al. 2017) which consists of three layers of full self attention including positional encoding to indicate the position of the node in the puzzle. The positional encoding consists of a fixed-size sine/cosine embedding of different frequencies, similar to Vaswani et al. (2017), but extended to 2D.

### Reward

Initial experiments were done using only a *sparse* reward: 0 reward to the agent if the game has not been solved and $R$ reward if the game is solved. As this reward provides very little feedback during training, agents struggle to learn effective strategies leading to slow learning and poor performance, especially in complex puzzles. Instead, we work with two types of reward systems: *iterative* and *partial*.

Given a graph $G$ with $n$ decision cells (not considering meta-nodes) can be encoded with a unique sequence $\Phi(G) = (g_i)_{i=1}^n$, with each $g_i \in \{1, 2, \ldots, m\}$ denoting the $i$-th value on the grid and $m$ representing the number of states a cell can take (e.g., $m = 4$ for Tents: empty, grass, tent, tree). The sequence corresponding to the puzzle's unique solution is denoted as $\hat{g} = \Phi(\hat{G})$, where $\hat{G}$ represents the solution graph. We measure the quality $Q(G)$ of a graph as its similarity to the solution:

$$Q(G) = \sum_{i=1}^{n} \delta(g_i, \hat{g}_i)$$

where $\delta(\cdot, \cdot)$ is the indicator function. To encourage policies to make iterative progress we follow the technique used by Tönshoff et al. (2022) which defines the reward $r^t$ based on the improvement in the graph's quality compared to the best quality $q^t = \max_{t' \leq t} Q(G^{t'})$ observed so far. The reward is then defined as the incremental improvement of the current state relative to the highest quality achieved in prior iterations:

$$r_{\text{iterative}}^t = \max \left(0, Q(G^{t+1}) - q^t\right)$$

This way, the agent earns a positive reward if the actions result in a state that is closer to the solution than any previous state. To avoid penalizing the policy for departing from a local maximum in search of superior solutions, we assign a reward of zero to states that are not an improvement.

The puzzles often encode violations of the puzzle rules explicitly, i.e. if two neighboring cells or all cells in a row violate a constraint. In the *partial reward* scheme, this information is given to the actors encoded in both the decision-nodes and the meta-nodes of the puzzle state. In the partial reward system, we adjust the calculation to only include nodes that are not part of a violation. For each node we have a indicator $C : \Phi(G) \to \{0, 1\}$ if it is part of such a violation. Then the quality of a graph and the corresponding reward is defined as:

$$\tilde{Q}(G) = \sum_{i=1}^{n} \delta(g_i, \hat{g}_i) \cdot C(g_i)$$

$$r_{\text{partial}}^t = \max \left(0, \tilde{Q}(G^{t+1}) - \tilde{q}^t\right)$$

The violation conditions are specific to the different puzzle environments. Thus, by incorporating puzzle-specific conditions into the reward calculation, the partial reward scheme aims to provide more meaningful and consistent feedback to the agent, promoting a more balanced and effective learning trajectory.

## Empirical Evaluation

### Comparison to Non-Graph Baselines

First, We compare our results to the baselines provided in the PUZZLES benchmark. Since the action space is different, we can only compare success rate but not episode length. The results are presented in Table 1. For the puzzles Tents, Mosaic, Loopy and Unruly, the GNN architecture was clearly able to surpass the performance of the baseline, even on much larger sizes. For Lightup and Net, the GNN architecture achieves similar performance to the baseline but on a substantially larger puzzle size. Note that all of these evaluations are done on the same size as the training distribution.

For the following evaluations, we report the interquartile mean performance of all puzzles, including 95% confidence intervals based on stratified bootstrapping, following (Agarwal et al. 2021).

### GNN and Transformer Baselines

Next, we compare the GNN architecture against a transformer baseline. The transformer baseline uses exactly the same nodes as the GNN. Because it misses the graph structure, the nodes are given a positional encoding. Our results in Figure 7 show that the GNN model performs much better than the Transformer baseline. We hypothesize that the GNN

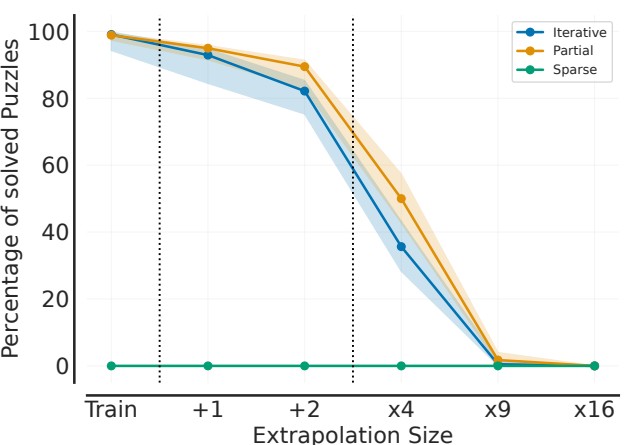

Figure 6: Percentage of puzzles solved during size extrapolation for models trained with different reward systems. During modest extrapolation, there seems to be no significant difference between the iterative and partial reward schemes. However, the partial reward scheme seems to allow for slightly better performance in the x4 and x9 extrapolation settings. Unfortunately, sparse rewards do not provide enough signal for the agents to learn any reasonable policy.

| Puzzle | Size | GNN Solved-% | Baseline Solved-% |
|--------|------|--------------|-------------------|
| Tents | 5x5 | $99.67 \pm 0.47\%$ | - |
|       | 4x4 | - | 45.0% |
| Lightup | 5x5 | $99.33 \pm 0.24\%$ | - |
|         | 3x3 | - | 99.1% |
| Mosaic | 4x4 | $100.0 \pm 0.0\%$ | - |
|        | 3x3 | - | 29.4% |
| Loopy | 4x4 | $68.83 \pm 7.17\%$ | - |
|       | 3x3 | - | 0% |
| Net | 4x4 | $99.83 \pm 0.24\%$ | - |
|     | 2x2 | - | 100.0% |
| Unruly | 6x6 | $83.67 \pm 19.01\%$ | 0% |

Table 1: Percentage of puzzles solved, average and standard deviation over all seeds, for the baseline GNN architecture compared to the best non-GNN architecture from (Estermann et al. 2024). Note that (Estermann et al. 2024) mostly trained agents on smaller and therefore much easier versions of the puzzles, using architectures unable to generalize to larger sizes.

can utilize the information about the explicitly encoded relational structure of the puzzle more effectively and as a result has a more suitable inductive bias compared to the transformer, helping it to better learn and then extrapolate on the puzzles.

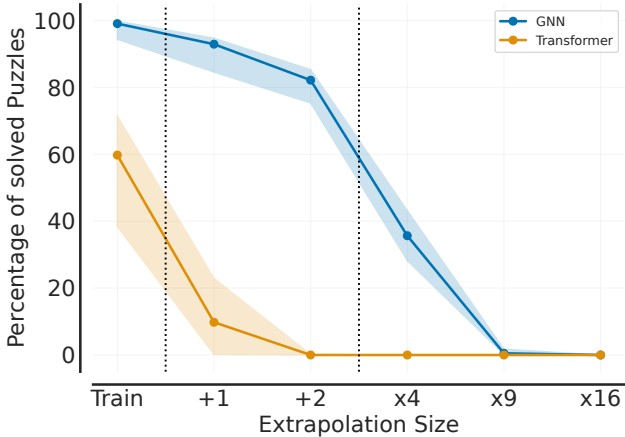

Figure 7: Compared to the transformer model, the graph architecture performs much better across the different puzzles, leading to a more consistent and successful extrapolation behavior. Both use the same modeling of the puzzles, but the GNN explicitly encodes the edge relationships, whereas the transformer is given a positional encoding.

### Recurrent vs State-less

Note that the puzzles are stateless, meaning that at every given moment during the solving, there is sufficient information to determine the solution with just the current puzzle state, i.e. what is visible on the board. However, it might still be beneficial to have a recurrent model for solving such a puzzle. Different steps towards computing the solution might require more computation or reasoning steps. As a consequence, for a GNN model it might be that the next correct action cannot always be determined with the information present in its 3 hop neighborhood. Therefore, it could be beneficial to allow these models access to a recurrent state, passing and storing information in between the different actions without affecting the game state. We compare a recurrent version of the graph architecture against a state-less variant in Figure 8. It seems that for modest extrapolation the recurrent version is more successful, whereas for larger sizes, the state-less architecture can solve more puzzles, even solving some instances that are 16 times larger.

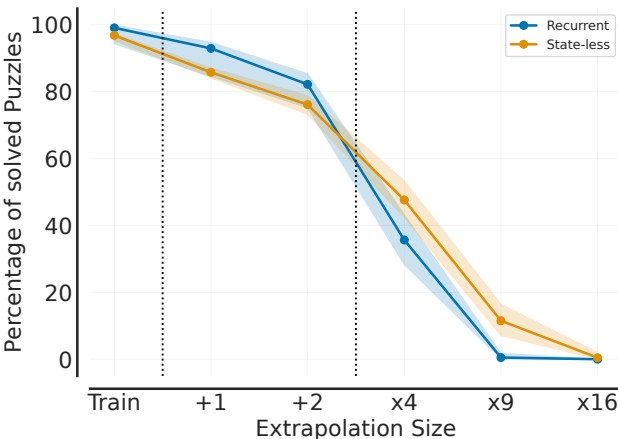

Figure 8: A recurrent agent design brings advantages for the success rate during training and modest extrapolation. For stronger extrapolation, however, a state-less algorithm performs better. The state-less algorithm is even able to solve a few puzzles at x16 extrapolation.

### Reward Systems

Finally, we want to evaluate the choice of reward system and its impact on size generalization. For this experiment, we use the GNN architecture with recurrent states. As depicted in Figure 6 by solely relying on sparse reward signals, it is very hard to learn to solve the puzzles, even during training. Both the iterative and partial variant show very comparable performance during training and modest extrapolation sizes. However, it seems that for the larger extrapolation setting the models trained using the partial reward generalize better.

## Limitations

Our study focuses on how to learn generalizable reasoning, which naturally imposes certain limitations. We deliberately concentrate on model-free reinforcement learning methods, excluding model-based approaches and pre-trained models including large language models. While these methods have their own advantages, our study wants to put the focus

on generalization from the given training distribution without relying on explicit given constructions or extending the training distribution itself.

Our research uses logic puzzles as a testbed for reasoning. While these provide an artificial environment with clear rules and scalability, they represent only a subset of reasoning tasks. Consequently, our findings may not directly generalize to all types of real-world reasoning problems or domains. Nevertheless, given the current challenges in developing truly generalizable reasoning systems, we believe that studying these techniques in controlled, synthetic environments is a crucial step towards advancing the field.

## Conclusion

The challenge of developing neural architectures capable of generalizable reasoning remains at the forefront of artificial intelligence research. Our study focuses on providing a controllable testbed of logic puzzles that can easily be scaled to test out-of-distribution generalization. Furthermore, through the unified graph representation, we demonstrate the potential of our graph-based multi-agent reinforcement learning approache in extrapolating their reasoning to larger, unseen instances.

In our empirical evaluation, we find that the graph-based modeling approach of the puzzles seems to be more fitting, resulting in both overall improved in-distribution and out-of-distribution performance compared to previous methods. Furthermore, we evaluate the inductive bias of a GNN architecture against a transformer baseline and find that the explicit graph structure aids generalization. Finally, we compare recurrent and state-less modeling for sequential decision making as well as different reward systems in the context of extrapolation. Our results underscore the challenges of achieving correct generalization without explicit guidance during training. This further highlights the importance of studying generalization in controlled environments. We aim to provide an stepping stone towards machine learning systems that can truly grasp the underlying reasoning and apply logical principles across diverse and increasingly complex domains.

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

## Puzzles

Simon Tatham's Portable Puzzle Collection features over 40 logic-based puzzles designed mainly to be little challenging and fun puzzles to pass time. The collection includes various types, from number placement and grid challenges to graph theory tasks. Each puzzle requires unique strategies involving deduction, pattern recognition, and planning. Their simple design and complex logic make them ideal for cognitive development and computational analysis. We focused on subset of these puzzles that are more suitable to be modeled as graph. Now we briefly recall the rules of the six puzzles implemented and tested.

### Tents

Tents is a logic puzzle where the objective is to place tents in a grid such that each tree has exactly one adjacent tent (horizontally or vertically), no two tents are adjacent to each other and the remaining cells are filled with grass. The numbers outside the grid indicate how many tents must be placed in each row or column.

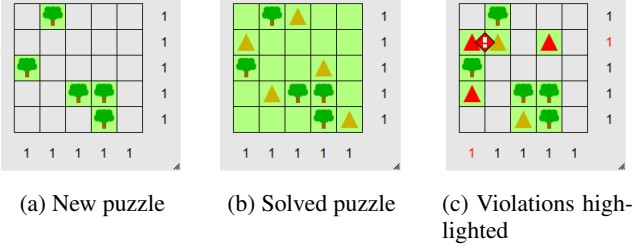

(a) New puzzle    (b) Solved puzzle    (c) Violations highlighted

Figure 9: Tents puzzle states: (a) New puzzle, (b) Solved puzzle, (c) Violations highlighted

### Lightup

Light Up is a grid-based logic puzzle where the objective is to illuminate all empty squares by placing light bulbs. Each bulb lights up its row and column until blocked by a black square. Numbered black squares specify the exact number

of adjacent bulbs required. Additionally, bulbs must not illuminate each other directly.

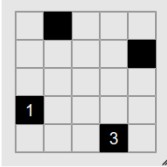 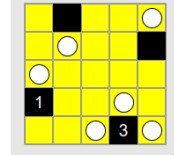 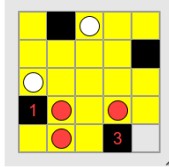

(a) New puzzle   (b) Solved puzzle   (c) Violations highlighted

Figure 10: Lightup puzzle states: (a) New puzzle, (b) Solved puzzle, (c) Violations highlighted

## Mosaic

Mosaic is a logic puzzle featuring a grid where each blue square must be colored black or white. The goal is to satisfy clue numbers that indicate the total number of black squares in the surrounding 3×3 region, including the clue square itself. The challenge is to strategically color the squares to meet all numerical constraints.

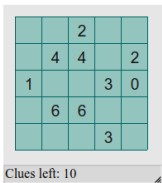 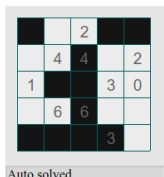 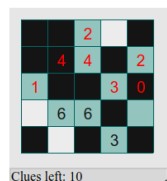

(a) New puzzle   (b) Solved puzzle   (c) Violations highlighted

Figure 11: Mosaic puzzle states: (a) New puzzle, (b) Solved puzzle, (c) Violations highlighted

## Loopy

Loopy is a logic puzzle where the goal is to draw a single continuous loop within a grid. The loop must pass bye the edge of some cells, adhering to numerical clues that indicate how many sides of the adjacent cells the loop must touch.

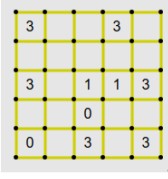 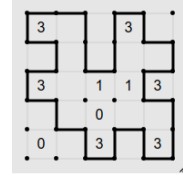 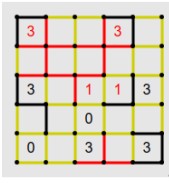

(a) New puzzle   (b) Solved puzzle   (c) Violations highlighted

Figure 12: Loopy puzzle states: (a) New puzzle, (b) Solved puzzle, (c) Violations highlighted

## Net

Net is a logic puzzle where the objective is to rotate tiles to connect all network pieces into a single, unbroken loop,

starting from an energy source tile. In particular in that game there aren't invalid moves

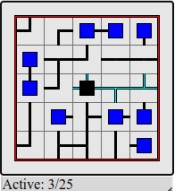 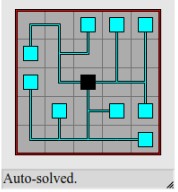 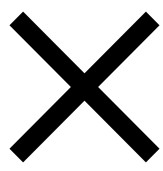

(a) New puzzle   (b) Solved puzzle   (c) Violations highlighted

Figure 13: Net puzzle states: (a) New puzzle, (b) Solved puzzle, (c) Violation highlighted (not present in Net).

## Unruly

Unruly is a logic puzzle where the objective is to color every square either black or white. The rules are: no three consecutive squares, horizontally or vertically, can be the same color, and each row and column must contain an equal number of black and white squares. Players left-click to turn squares black, right-click to turn them white, and middle-click to reset them to empty.

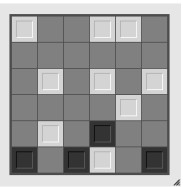 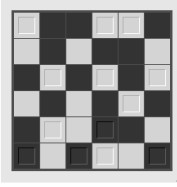 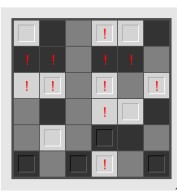

(a) New puzzle   (b) Solved puzzle   (c) Violations highlighted

Figure 14: Unruly puzzle states: (a) New puzzle, (b) Solved puzzle, (c) Violations highlighted.

## Modeling Puzzles as Graphs

The exact topology of the graph and the node attributes for Net, Loopy, Mosaic and Tents can seen in figures [15, 16, 17, 18].

Unruly has the same topology as tents. The node attributes include:

- An indicator of whether the cell is white or black.
- A flag indicating whether the block is fixed.
- A flag indicating whether the node is a metanode
- Three flags for violations: horizontal, vertical, and number violations.

Horizontal and vertical violations occur when a cell is part of a horizontal or vertical strip (or both) of the same color with at least three elements. The number constraint controls whether there is an imbalance between white and black squares in a row or column.

Lightup has the same topology as net, meaning there are no metanodes. The node attributes include:

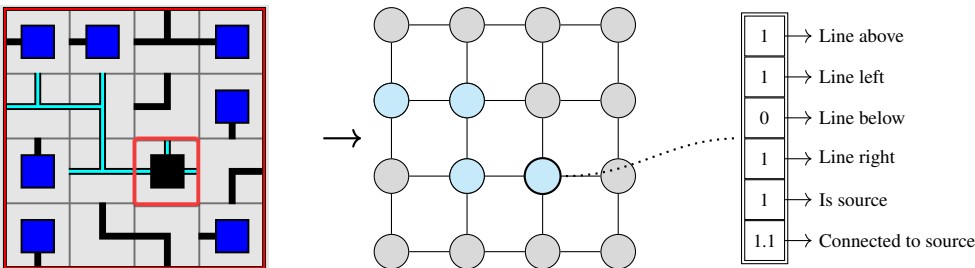

Figure 15: Illustration of the graph and node attributes used for net. This game (together with lightup) presents the simplest topology: each node is a cell of the grid of the game and the edges are only between vertically or horizontally adjacent cells.

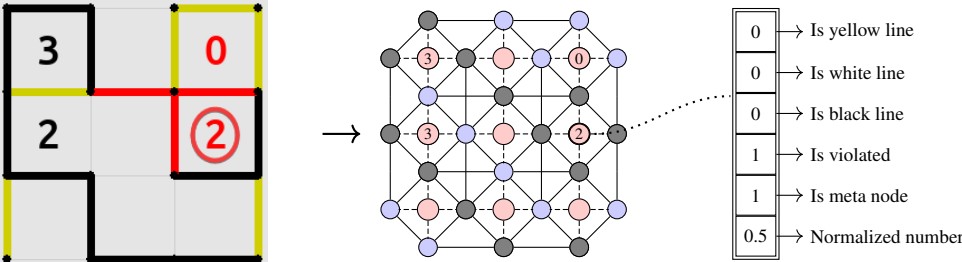

Figure 16: Illustration of the graph and node attributes used for loopy. In this case each decision node (black or white circles) corresponds to an edge of the game grid. For each square of the grid we have a meta-node (red circle) connected to its 4 surrounding decision nodes.

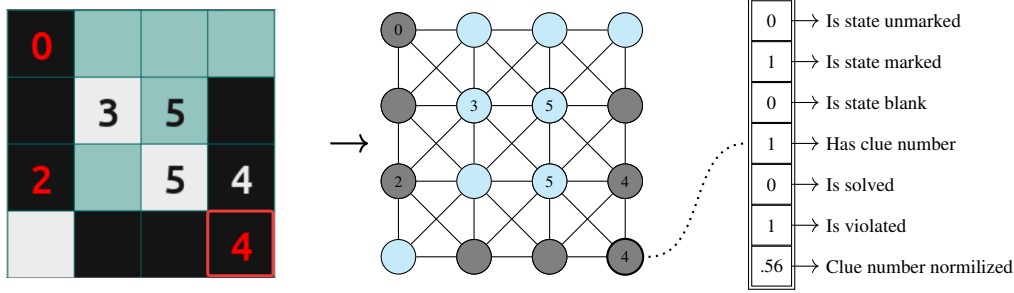

Figure 17: Illustration of the graph and node attributes used for mosaic. Here, differently from the net the nodes are cells that are diagonally aligned are also connected together in the graph. This is because, the game itself requires checking for all neighbors (including diagonal neighbors) for each cell with a clue number.

- An indicator of whether the cell is a bulb, is empty, is lighted (yellow), is a black (fixed) square, or a black square with a number.

- A flag for violation.

- The normalized number of the block (non-zero if it has one).

### Action Spaces

In our graph based modeling of the puzzle, at each step, every cell can choose an action to modify the current state of the puzzle. This approach contrasts with a previously implemented cursor model (Estermann et al. 2024), which was found to perform significantly worse. The cursor model operates differently, providing only a single action at each timestep. The cursor can either change the state of the cell it is positioned on or move to another adjacent cell through

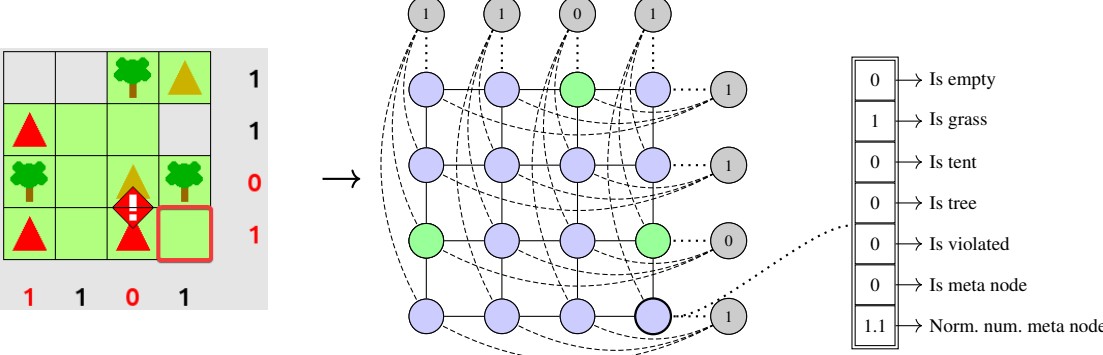

Figure 18: Illustration of the graph and node attributes used for tents. In this case, we have the decision nodes (blue and green circles) that are structured in the same way as net's nodes are. Moreover, we add meta-nodes (black circles) that represent constraints on the number of tents to be placed in each row or column.

actions like left, right, up or down.

| Action | Net | Lightup | Loopy | Tents | Mosaic | Unruly |
|---|---|---|---|---|---|---|
| 1 | Rotate 90° | Place lightbulb | Mark line | Place tent | Mark cell | Turn cell white |
| 2 | Rotate 180° | Empty Cell | Unmark line | Place grass | Unmark cell | Turn cell black |
| 3 | Rotate 270° | DO NOTHING | Empty line | Empty cell | Empty cell | DO NOTHING |
| 4 | DO NOTHING | - | DO NOTHING | DO NOTHING | DO NOTHING | - |

Table 2: Possible actions for each puzzle type.

Each decision node in the graph represents a possible action in the game. For each puzzle, we have a distinct set of actions associated with every decision node. Actions decided for the meta-nodes are discarded, as meta-nodes are used to convey shared information rather than direct actions. The final action for each node is selected using a softmax layer in the GNN. Once the actions are chosen for all nodes, they are executed simultaneously, resulting in a new state of the game. We illustrate the specific actions associated to a single cell of each game in Table 2.

### Determining Puzzle Dimensions

Initial experiments have shown that if the puzzle size is chosen too small, the models tend towards overfitting on the smaller-sized games and not generalize at all. This is likely linked to the fact that on small puzzle sizes the number of different puzzle configurations is quite limited. Therefore, it is possible to more easily overfit to the training data. On the other hand, we want to avoid very large puzzle sizes during training as they often require more steps in order to be solved which is both a burden for the models to learn and computationally more expensive. Therefore, we decide to determine for each puzzle the smallest size which allows for at least 40'000 unique puzzle configurations. The final training sizes for each puzzle are highlighted in Table 3.

## Empirical Evaluation

As highlighted previously, our objective is to train a reinforcement learning (RL) agent capable of solving a subset

| Size | Tents | Lightup | Mosaic | Loopy | Net | Unruly |
|---|---|---|---|---|---|---|
| 3 × 3 | - | - | - | 573 | 176 | - |
| 4 × 4 | 240 | 5'864 | 872 | **85'785** | 42'029 | - |
| 5 × 5 | **85'118** | **412'624** | 192'533 | 468'671 | 500'000+ | - |
| 6 × 6 | 494'248 | 499'883 | 499'680 | 500'000+ | 500'000+ | **41'222** |
| 7 × 7 | 500'000+ | 500'000+ | 500'000+ | 500'000+ | 500'000+ | 500'000+ |

Table 3: Number of distinct puzzle configurations for each puzzle depending on the puzzle size. The shown number of puzzles is a lower bound on the actual number, which was estimated using 500'000 randomly sampled instances.

| Puzzle | Tents | Lightup | Mosaic | Loopy | Net | Unruly |
|---|---|---|---|---|---|---|
| Training | 5x5 | 6x6 | 4x4 | 4x4 | 4x4 | 6x6 |
| Validation | 6x6 | 6x6 | 5x5 | 5x5 | 5x5 | 8x8 |
| +1 | 6x6 | 6x6 | 5x5 | 5x5 | 5x5 | 8x8 |
| +2 | 7x7 | 7x7 | 6x6 | 6x6 | 6x6 | 10x10 |
| x4 | 10x10 | 10x10 | 8x8 | 8x8 | 8x8 | 12x12 |
| x9 | 15x15 | 15x15 | 12x12 | 12x12 | 12x12 | 18x18 |
| x16 | 20x20 | 20x20 | 16x16 | 16x16 | 16x16 | 24x24 |

Table 4: Puzzles sizes used for training, validation and testing.

of Simon Tatham's puzzles using a Graph Neural Network (GNN) for both policy and value estimation. The core of the training loop involves executing the Proximal Policy Optimization (PPO) algorithm.

In particular each iteration of PPO consists of two key phases: the rollout phase and the update phase. During the rollout phase, experience is gathered in the form of `<batch_size>` tuples containing *(State, Action, NextState)* pairs. This experience is stored in a buffer, which is then used in the subsequent update phase. In the update phase, the parameters of the networks (critic and actor) are adjusted. This is done by looping through the collected experiences in the rollout buffer over a number of `<epochs>`.

A puzzle is considered solved in an iteration if the number of `steps` required to find a solution is less than or equal to the predefined `<horizon>`. Otherwise, the game is reset to a new state, and the step count restarts, allowing the agent to attempt solving the puzzle from a fresh configuration.

## Training Parameters and Baseline Establishment

In addition to selecting the optimal puzzle dimensions, a crucial aspect of training the reinforcement learning (RL) algorithms involved fine-tuning several key parameters. This process ensured that the models were robust, generalized well across different puzzle sizes, and performed efficiently.

**Tunable Parameters**   The parameters that were subject to tuning played a significant role in the performance and generalization capabilities of the RL models. These parameters were carefully selected based on initial experiments, and their values were adjusted iteratively to identify the best-performing configurations. Below is a list of the tunable parameters, along with the values that were considered during the training phase:

1. `reward_mode` → [*sparse*, ***iterative***, *partial*]. The two different reward schemes that were explained in the reward section.

2. `glbh` → [***recurrent***, *state-less*]. The recurrent and state-less variants of the architecture.

* `net_arch` → [***gcn***, *transformer*]. The `gcn` flag represents the GENConv architecture of the GNN while `transformer` represents the transformer architecture.

The parameters in bold represent the baseline configuration that was used as a starting point for all experiments. This baseline provided a reference model against which other configurations were compared.

**Sequential Tuning Process**   The training procedure followed a sequential tuning approach to manage the complexity of simultaneously testing multiple parameters, which would otherwise lead to an exponential increase in combinations. For each game and each GNN architecture, we started with the baseline configuration and then systematically tuned one parameter at a time.

**Fixed Parameters**   While certain parameters were adjustable, others were kept constant throughout the training process to ensure consistency and comparability across experiments. Some notable ones are:

- `timesteps`: Number of timesteps to reach for the algorithm to stop learning. For the GNN, set to 2,000,000 for each game, with exceptions for 'loopy' (2,400,000) and 'tents' (1,000,000). While for Transformer set to 2,000,000 for each game, with exceptions for 'loopy' (2,400,000).

- `batch_size`: Fixed at 320 for GNN while is equal to 3200 for Transformer, with a minibatch size of 32 for both.

- `lr`: Learning rate used, set to 0.0003 for GNN and 0.00006 for Transformer with the Adam optimizer.

- `mp_arch`: The pooling function used in the message passing layers. Set to "mean function".

- `layers`: Number of message passing layers. Set to 3 for both architectures.

- `hidden_dim`: Size of the hidden dimension for node and edge embeddings after the network's encoder. Set to 32.

- `ent_coef`: Entropy coefficient for PPO, set to 0.004 for both architectures.

- `gamma`: Gamma parameter for PPO, set to 0.5 for both architectures.

These fixed parameters were chosen based on preliminary tests that indicated they provided a good balance between training stability and computational efficiency.