# OpenReview forum: "Beyond Interpolation: Extrapolative Reasoning with Reinforcement Learning and Graph Neural Networks"
_AAAI.org/2025/Workshop/NeurMAD — AAAI 2025 Workshop NeurMAD Submission_

### Official Review · Reviewer_Z4KK · 2024-12-18

**Rating:** 7
**Confidence:** 3

**Review:**

#### Summary
This paper tackles the challenge of enabling neural architectures to generalize beyond their training distributions.
 By employing logic puzzles as a controlled testbed, the authors propose a novel graph-based approach coupled with reinforcement learning (RL) to model scalable logical structures.
The key contributions include a multi-agent RL framework leveraging Graph Neural Networks (GNNs) for reasoning, insights into the role of architectural inductive biases, reward system designs, and recurrent modeling in achieving extrapolative reasoning.

---

#### Strengths

1. The paper effectively identifies generalization beyond interpolation as a critical challenge in machine learning. Using logic puzzles as a testbed provides a well-defined, scalable, and interpretable framework.

6. The introduction of the PUZZLES benchmark and a graph-based interface for logic puzzles enriches the resources available for studying generalization in controlled environments.

2.  Representing puzzles as graphs provides a flexible and scalable way to handle tasks of varying complexity. The dual use of decision and meta-nodes in GNNs captures local and global constraints effectively.

3. The experiments are comprehensive, spanning multiple puzzle types, varying sizes, and different architectural choices (GNNs vs. transformers). The evaluation metrics (e.g., extrapolation to x4, x9, x16 larger puzzles) are robust and directly aligned with the paper's goals.

---

### Official Review · Reviewer_DeFn · 2024-12-28
**A long full paper, lack of key information**

**Rating:** 4
**Confidence:** 4

**Review:**

Summary

This paper integrates GNN (graph neural network) with RL (reinforcement learning) to address PUZZLES problems. This paper claims the GNN can improve RL performance.

Compared with RL-only method, the new method reports considerable improvement on small size puzzles (2x2 ~ 6x6), but it only reports similar performance (to RL-only method) on larger size puzzles.

Beside above comparison, other empirical evaluations are based on the new method itself with different configurations, e.g.: iterative vs partial, recurrent vs state-less, GNN vs transformer. All these tests show a complete zero-solvability (cannot be solved) at x9 scale size. This empirical result matches the conclusion from other methods and research.

Issues

1. this paper does not provide a comprehensive explanation of how these puzzles transformed to a GNN network. Also, no runable program provided.

2. At large scale-size problem, this GNN+RL method matches the performance of RL-only method, it only perform better at (2x2 ~ 6x6) small scale size. Also, it lacks enough cross-comparison from RL-only method.

3. The major reason for its better performance (at small scale) is due to its richer NN-node representation capturing more state relationships. This benefit is not scalable when PUZZLES size is growing up.

---

### Official Review · Reviewer_tUXR · 2024-12-29
**a very interesting approach**

**Rating:** 7
**Confidence:** 1

**Review:**

this is a very interesting approach to solve puzzles. It uses RL and graph neural networks. However it is outside my area of expertise.

It is a well written paper and the ideas are well presented. I would like my review to be moderated based on the fact that I am not an expert in this field.

---

### Decision · Program_Chairs · 2024-12-30

**Decision:**

Accept

**Comment:**

 We agree with the major opinions of reviewers and authors’ comments.